# Protein Arginine Methyltransferases in Pancreatic Ductal Adenocarcinoma: New Molecular Targets for Therapy

**DOI:** 10.3390/ijms25073958

**Published:** 2024-04-02

**Authors:** Kritisha Bhandari, Wei-Qun Ding

**Affiliations:** Department of Pathology, University of Oklahoma Health Sciences Center, BMSB401A, 940 Stanton L. Young Blvd., Oklahoma City, OK 73104, USA; kritisha-bhandari@ouhsc.edu

**Keywords:** pancreatic ductal adenocarcinoma (PDAC), arginine methylation, protein arginine methyltransferases (PRMTs), molecular targets

## Abstract

Pancreatic ductal adenocarcinoma (PDAC) is a lethal malignant disease with a low 5-year overall survival rate. It is the third-leading cause of cancer-related deaths in the United States. The lack of robust therapeutics, absence of effective biomarkers for early detection, and aggressive nature of the tumor contribute to the high mortality rate of PDAC. Notably, the outcomes of recent immunotherapy and targeted therapy against PDAC remain unsatisfactory, indicating the need for novel therapeutic strategies. One of the newly described molecular features of PDAC is the altered expression of protein arginine methyltransferases (PRMTs). PRMTs are a group of enzymes known to methylate arginine residues in both histone and non-histone proteins, thereby mediating cellular homeostasis in biological systems. Some of the PRMT enzymes are known to be overexpressed in PDAC that promotes tumor progression and chemo-resistance via regulating gene transcription, cellular metabolic processes, RNA metabolism, and epithelial mesenchymal transition (EMT). Small-molecule inhibitors of PRMTs are currently under clinical trials and can potentially become a new generation of anti-cancer drugs. This review aims to provide an overview of the current understanding of PRMTs in PDAC, focusing on their pathological roles and their potential as new therapeutic targets.

## 1. Introduction: Pancreatic Ductal Adenocarcinoma and the Need for New Therapeutics

Pancreatic ductal adenocarcinoma (PDAC) is a major type of pancreatic neoplasm that originates from ductal or acinar cells, comprising more than 90% of pancreatic cancer cases [1]. It is the third-leading cause of cancer-related death in the United States and is predicted to surpass colorectal cancer by 2040, to become the second-leading cause of cancer-related death [2]. The current 5-year overall survival rate for this disease is 13%, which is lower than that for most solid tumor types [3]. Approximately 80–85% of patients are diagnosed with PDAC when the disease has already metastasized or became locally advanced, making them ineligible for surgical resection [4,5,6,7]. For the remaining 15–20% of PDAC patients that are diagnosed early and are eligible for surgical resection, 3 out of 4 patients will develop a relapse within 2 years post-operation [7,8]. In both cases, whether considering patients as surgical candidates or patients with metastatic disease, they usually all undergo intensive chemotherapy. The first line of treatment includes the use of two different regimens: FOLFIRINOX, which is the combination of folinic acid (leucovorin), 5-fluorouracil (5-FU), irinotecan, and oxaliplatin, or gemcitabine combined with nab-paclitaxel [8,9]. The second line of treatment includes liposomal formulation of Irinotecan with 5-Fluorouracil. Patients are eligible to switch to the second line of treatment if their disease progresses during the first line of treatment and they have not received these second-line drugs previously [8].

The major challenge with current PDAC chemotherapy is the development of drug resistance, which has mostly been observed in gemcitabine-treated patients [10]. In addition, the combination of several drugs in FOLFIRINOX is extremely toxic and has a severe impact on the patient’s quality of life [11]. Unfortunately, current molecular targeted therapy and immunotherapy, which have shown unprecedented therapeutic benefits for other cancer types, are rarely effective for patients with PDAC [12,13,14,15]. The only FDA-approved targeted therapeutic for PDAC is the epidermal growth factor receptor inhibitor erlotinib, which slightly prolongs patient survival [13]. Currently available immunotherapies have shown limited efficacy in improving PDAC patient survival [12]. New strategies in developing effective therapeutics against PDAC are desperately needed. 

It should be noted that decades of research in the biology of PDAC have led to the discovery of many promising molecular targets for this disease, such as the KRAS mutation that leads to activation of oncogenic signaling, the desmoplastic tumor microenvironment (TME) that facilitates immune evasion, and the altered tumor metabolism that contributes to chemo-resistance [8]. The potential therapeutic benefit of targeting the mutated KRAS protein in PDAC has been extensively explored, as more than 90% of patients with PDAC harbor this mutation [4,8,16]. However, while the mutated KRAS protein is a “druggable” target when using KRAS specific small-molecule inhibitors (there are no FDA-approved KRAS inhibitors for PDAC as of yet), PDAC cells often find a way to adapt by following an “RAS independent” pathway, compromising the efficacy of the small-molecule inhibitors [17,18,19,20].

The altered tumor metabolism for PDAC provides a wide window of opportunities to develop new therapeutic interventions. Targeting autophagy, glutamine metabolism, and glycolysis through lactate dehydrogenase inhibition, in combination with other chemotherapeutics, has been experimentally explored [8,21,22], but clinically applicable therapeutics that target tumor metabolism have not been available for PDAC. Targeting the extracellular matrix (ECM) barrier (the desmoplastic TME) has been expected to improve drug access to the tumor tissues but has not been successful in clinical trials [23,24]. Immunotherapies based on the development of cancer vaccines, checkpoint inhibitors, CAR T-cells, and stroma and myeloid targeting remain ongoing in clinical trials for PDAC [25]; however, it is known that most previous clinical trials on PDAC therapeutics have fallen short of expectations [8]. New molecular targets and therapeutic strategies for PDAC merit further exploration.

Recent advances in our understanding of the biology of PDAC has demonstrated one group of the promising candidate therapeutic targets in PDAC cells: a family of enzymes named protein arginine methyltransferases (PRMTs). PRMTs are enzymes that are responsible for methylating arginine residues in histone as well as non-histone proteins [26]. They play a multitudinous role in the biology of cells and are associated with the progression of diseases, such as cancer. Expression of several PRMTs is upregulated in PDAC cells and tissues, thereby promoting progression of the disease [27]. Drugs targeting PRMTs are known to be efficacious in killing cancer cells in vitro as well as in vivo, either alone or in combination with chemotherapeutic agents. Some of the inhibitors of PRMTs are currently in clinical trials [28]. This review introduces the family of PRMTs in mammalian cells, their involvement in the pathogenesis of PDAC, and the potential of PRMTs as therapeutic targets for PDAC.

## 2. Arginine Methylation and PRMTs

Protein methylation is the fifth-most abundant post-translational modification (PTM) and is observed in histone as well as non-histone proteins [29]. While several amino acids are known to undergo this modification, the major amino acids that are known to be methylated are lysine and arginine [30]. The lysine methylation in histone and non-histone proteins has been extensively reviewed elsewhere [31,32,33]. Therefore, this review focuses on arginine methylation of these proteins. PRMTs are the enzymes that catalyze arginine methylation, which is ubiquitously present in both the nuclear as well as cytosolic compartments of the cells [34]. These enzymes can be broadly classified into three different categories: Type I, Type II, and Type III, based on their ability to catalyze different modes of arginine methylation in the proteins. While Type I PRMTs can form mono-methyl arginine (MMA) and asymmetric di-methyl arginine (ADMA), Type II PRMTs produce mono-methyl arginine (MMA) and symmetric di-methyl arginine (SDMA), and Type III PRMTs can only produce mono-methyl arginine (MMA) (Figure 1) [26]. PRMT1, PRMT2, PRMT3, PRMT4/CRAM1, PRMT6, and PRMT8 are Type I PRMTs; PRMT5 and PRMT9 are Type II; and PRMT7 is a solo Type III PRMT. The method by which PRMTs methylate arginine residues has been well described. PRMTs utilize cofactor S-adenosylmethionine (SAM, also known as AdoMet) to catalyze the transfer of methyl groups to the guanidino nitrogen moieties in arginine residues of the substrate protein. This reaction yields the formation of methylarginine, with S-adenosylhomocysteine (SAH) as the side product [35]. All the PRMTs are expressed in the pancreas, with the exception of PRMT8, whose expression is known to be limited to the brain [36]. The expression and localization of different types of PRMTs vary between the endocrine and exocrine regions of the pancreas. Detected by enzyme-specific antibodies, PRMTs are ubiquitously expressed in either the islets of Langerhans or pancreatic acini, with differential expression across cellular compartments, as shown in Table 1.

### 2.1. Structural Basis, Localization, and Motif Preference of PRMTs

The canonical structure of PRMTs contains three structural domains: an N-terminal catalytic core (Rossman fold), the α-helical dimerization arm, and the C-terminal ß-barrel domain (Figure 2) [37]. The catalytic core in PRMTs consists of approximately 300 amino acids, containing the SAM-binding site. The ß-barrel domain facilitates the binding of substrates to PRMTs. The dimerization arm is essential in most PRMTs, as PRMTs function as dimers, with the exception of PRMT7 and PRMT9 [38,39]. The structural features of individual PRMTs are explained in detail in Table 2. PRMTs are localized between different cellular compartments and have their own motif of preference for arginine methylation (Table 3). While there is no unanimous recognition motif for different types of PRMTs, scientific findings suggest that the glycine-rich motifs, such as RGG, RxR, and GAR, have a high likelihood of being methylated by PRMTs [34,40]. One recent finding indicates that PRMT5 recognizes a GRG motif to methylate its substrate [41]. Most of the current studies utilize the arginine methylation prediction tools PRmePRed (https://bioinfo.icgeb.res.in/PRmePRed/, accessed on 20 February 2024) [42] or GPS-MSP (http://msp.biocuckoo.org/online.php, accessed on 20 February 2024) [43] to gain a tentative idea of arginine methylation sites in the protein of interest, and then validate these sites using in vitro arginine methylation assays combined with site-directed mutagenesis and proteomics. Interestingly, all of the PRMTs are also highly regulated by different PTMs that may or may not affect their catalytic activity [44,45,46,47,48,49,50,51,52].

The mechanism of arginine methylation by PRMTs has been extensively studied with SAM as a methyl donor to methylate their substrates [35]. The substrates of PRMTs are present in both the nucleus as well as the cytoplasm. While there has been a debate about the kinetics of these enzymes, most findings suggest a multi-step methylation of the substrate, also called a distributive process, wherein the substrate is first mono-methylated, followed by its dissociation from the SAM–PRMT–substrate complex and a di-methylation of the substrate by re-formation of the donor–enzyme–substrate complex [35,60,61,62].

### 2.2. Physiological Role of PRMTs

The arginine methylation of proteins, like any other types of PTMs, increases the diversity of cellular proteome and, therefore, plays a significant role in maintaining cellular homeostasis. The addition of a methyl group in the arginine residues does not seem to alter the charge of the protein; rather, it can facilitate or disrupt the interaction among proteins and nucleic acids, resulting in diverse physiological responses [37]. Given the abundance of protein arginine methylation in eukaryotic cells, there is no doubt that PRMTs are involved in many aspects of cellular function. Studies have revealed that the PRMT substrates are mostly associated with RNAs [63]. Not surprisingly, PRMTs have been shown to mediate gene transcription [64], mRNA splicing [65,66], DNA damage repair (DDR) [67,68], cell stemness [35], etc.

PRMT1 and CARM1 are known to act as transcription coactivators via histone arginine methylation, which facilitates the binding of transcription factors, such as ERα [63,69], p53, YY1 [70], and PPARγ [71], to the promoter of genes. Furthermore, PRMT1 is shown to be able to methylate arginine residues in non-histone proteins to co-activate gene transcription [72]. PRMT1-mediated transcription activation enhances EGFR signaling and promotes colorectal cancer progression [73]. On the other hand, PRMT5 and PRMT6 are shown to have transcription co-repressor activity that suppresses gene transcription via arginine methylation of histone proteins [34]. PRMT5 is considered a general transcription repressor via arginine methylation that interacts with different transcription factors or repressor complexes, such as Snail [74] and BRG1 [75], to repress gene transcription. PRMT5-mediated arginine methylation of histone proteins can repress expression of epithelial junctional genes, thereby promoting cancer cell invasion [76]. However, PRMT5 is also reported to potentiate gene transcription through arginine methylation of histone proteins that enhances the binding of a transcriptional co-activator [77]. Other PRMTs, including PRMT2 and PRMT7, are also reported to mediate gene transcription through arginine methylation of histone proteins [34]. These findings indicate that PRMTs play a critical role in regulating gene transcription through arginine methylation in a context-dependent manner.

The regulation of mRNA splicing is a well-known function of PRMTs, especially CARM1 and PRMT5 [78]. One of the well-explored examples is the arginine methylation of Sm proteins in the GAR motif towards its C-terminal domain by PRMT5, facilitating its interactions with the Tudor domain of the survival of motor neuron (SMN), which plays a key role in maintaining the fidelity of constitutive nuclear splicing events and RNA metabolism. The deletion or inhibition of PRMT5 reduces spliceosome assembly and causes aberrant splicing, such as intron retention [79]. PRMT5 has also been reported to regulate mRNA splicing of the p53 inhibitor MDM4 [66]. Upon deletion of PRMT5, there is a formation of shorter and less stable MDM4 isoform protein due to alternative splicing of the transcript, which is incapable of inhibiting p53. CARM1 has been involved in the regulation of mRNA splicing via methylating RBPs [80]. Direct methylation of splicing factors by CARM1, thereby affecting the RNA spicing process, has been well described [81], and, similar to PRMT5, CARM1 also causes Sm protein methylation in a model system [81]. In contrast to CARM1 and PRMT5, the involvement of PRMT1 in mRNA splicing has been elusive. Specific substrates by which PRMT1 may regulate RNA splicing have not been identified. However, PRMT1 is known to regulate RBP cellular localization [82], and depletion of PRMT1 is associated with aberrant RNA splicing in mouse cardiomyocytes [83].

With regard to DNA damage response (DDR), PRMT5 and PRMT1 have been shown to be critical for the repair of the damaged DNA. PRMT1-null mouse embryonic fibroblasts exhibit genome instability, spontaneous DNA damage, and checkpoint defects [84]. These are mediated by MRE11, an integral component of the MRN complex known to activate the DDR pathways. PRMT1-mediated arginine methylation of MRE11 helps it anchor to the DNA double-strand breaks (DSBs), stimulating nuclease activity, while its demethylated version MRE11^RK^ is defective in DNA end-resection and ATR activation [85]. In the case of PRMT5, it is known that PRMT5 deficiency causes spontaneous DNA damage and defects in homologue recombination-mediated DSB repair [86,87]. PRMT5 stabilizes RPA2, which is one of the three subunits of the RPA complex. The RPA complex binds and protects ssDNA formed during DNA repair. Knockout of PRMT5 results in the depletion of RPA2, causing RPA exhaustion. This leads to impaired homology-directed repair (HDR) of the cells treated with gemcitabine, thereby enhancing the efficacy of gemcitabine treatment in pancreatic cancer cells [68]. PRMT5-mediated arginine methylation has also been shown to regulate several other proteins that are involved in the DDR process, including p53-binding protein 1 [88], RAD9 [89], RUVBL1 [86], and TDP1 [90]. Other PRMTs, such as CARM1 and PRMT6, are also directly or indirectly involved in regulating the DDR process through protein arginine methylation [91,92].

Other biological functions of PRMTs include serving as mediators to promote apoptosis [54], participating in regulating tumor immunity [93,94], and governing stem cell fate and survival during embryogenesis as well as adult homeostasis [35].

## 3. PRMTs Are Involved in the Pathogenesis and Progression of PDAC

Given the broad implication of PRMTs in the biology of eukaryotic cells, the involvement of PRMTs in the pathogenesis and progression of PDAC is not a surprise. It has been found that some of the PRMTs, especially PRMT1, PRMT3, and PRMT5, are intimately associated with PDAC tumorigenesis, metastasis, and chemo-resistance, as discussed below.

### 3.1. PRMT1

PRMT1 accounts for about 85% of the total Type I PRMT activities and is responsible for the MMA or ADMA of both histone and non-histone proteins [95]. The consequence of arginine methylation by PRMT1 is observed in the processes of transcription regulation, signal transduction, or DNA damage repair [26]. PRMT1 is known to be involved in the tumorigenesis of different cancer types. For example, it affects several signaling cascades associated with the hormonal receptors for estrogen and progesterone in breast cancer cells [96,97]. Its involvement in other cancer types, such as lung cancer [98] and colorectal cancer [99], is also reported. Notably, expression of PRMT1 is upregulated in PDAC tissues. Tissue microarray by immunohistochemistry involving tissue samples of 90 patients has shown an overexpression of PRMT1 in PDAC tissues compared to the adjacent normal pancreatic tissues. PRMT1 expression level in PDAC is found to be positively correlated with the tumor size and post-operative patient prognosis [100]. In an experimental setting, PRMT1 promotes growth of pancreatic cancer cells in vitro and in vivo via enhancing the ß-catenin level [100]. Individual substrates for PRMT1 that are involved in PDAC progression or chemo-resistance have been described. For example, Gli1, an oncogenic transcription factor, essential for Hedgehog signaling, is a substrate of PRMT1. Gli1 is methylated at the R597 residue, which is critical for its transcriptional activity, and its demethylation sensitizes PDAC cells to gemcitabine [101]. The expression of methylated Gli1 positively correlates with PRMT1 expression, suggesting that PRMT1 methylates Gli1, thereby enhancing its oncogenic activity and promoting PDAC progression [101]. It is also observed that overexpression of PRMT1 in PDAC cells facilitates the arginine methylation of HSP70, which aids in the stabilization of BCL2 mRNA, augmenting the expression of BCL2 protein, which prevents cellular apoptosis and renders PDAC cells chemo-resistant [102]. A recent study demonstrates that PDAC disease maintenance depends on PRMT1-mediated RNA metabolism and cellular processes, further indicating that PRMT1 is a therapeutic target for PDAC [103]. In this study, an RNAi-based screening using patient-derived PDAC cells identifies PRMT1 as a top epigenetic lethality factor. Both knockdown and pharmacological inhibition of PRMT1 in PATC53 cells (PDAC patient-derived cells) results in the reduction of cell proliferation and colony formation in vitro as well as reduced tumor volume in vivo, with a significant decrease in cellular and tissue ADMA levels and an increase in the MMA levels. Knockdown of other Type I PRMTs, including PRMT4 and PRMT6 in the same PDAC model system, has no such effects, indicating the critical dependency of PDAC on PRMT1. Proteomics and transcriptomic analysis upon pharmacological inhibition of PRMT1 reveal that this dependency is most likely due to PRMT1-mediated RNA metabolism, cell cycling, DNA replication, and DNA repair in patient-derived PDAC cells [103]. Specifically, pharmacological inhibition of PRMT1 downregulates expression of genes associated with cell cycle, DNA replication and DNA repair. The binding and methylation of RBPs by PRMT1 regulates RNA splicing, 3′-end RNA processing and RNA stability, and protein translation efficiency [103].

As stated earlier, immunotherapies are less effective in patients with PDAC [12]. However, the poor therapeutic response of immune checkpoint inhibitors that target programmed death ligand-1 (PD-L1) in PDAC is somewhat rescued by co-administration with the PRMT1 inhibitor PT1001B. PT1001B, in conjunction with an anti-PD-L1 antibody, enhances the inhibition of PD-L1 expression in tumor cells and the infiltration of CD8+ lymphocytes, and reduces PD-1+ leukocytes in vivo, thereby augmenting the efficacy of the immune checkpoint inhibitor in a PDAC model system. These observations support the notion that PRMT1 is a molecular target through which the efficacy of the anti-PD-L1 therapy can be enhanced in PDAC [104].

PRMT1 is also known to regulate the EMT-signaling pathway in cancer cells, suggestive of its involvement in tumor metastasis. The expression of one of the key components of the EMT pathway, ZEB1, is highly associated with the expression of PRMT1 in PDAC cells [105]. Downregulation of PRMT1 in PANC-1 and SW1990 cells reduces cell proliferation and invasion, yet overexpression of PRMT1 did not affect these events, which is explained by the high level of endogenous PRMT1, leading to saturation of the PRMT1 protein in pancreatic cancer cells. The anti-tumor effect of PRMT1 downregulation is reversed by overexpression of ZEB-1, indicating the role of PRMT1–ZEB1-signaling cascade in pancreatic cancer progression [105]. The role of PRMT1 in promoting invasion or metastasis of other type of cancers is also well recognized [106,107,108].

### 3.2. PRMT3

PRMT3 is a Type I PRMT. Expression level of PRMT3 is correlated with patient prognosis for PDAC, liver cancer, colorectal cancer, and prostate cancer (https://www.proteinatlas.org accessed on 19 February 2024). Overexpression of PRMT3 is evident in PDAC tissues and is associated with poor survival in PDAC patients [109]. While studies of PRMT3 focus more on the control of apoptosis and tumor progression in breast cancer [110,111], its involvement in PDAC has been mainly associated with chemo-resistance [112], likely due to PRMT3-mediated metabolic reprograming of PDAC cells [109]. Studies have shown that PRMT3 upregulates expression of the multidrug-resistant gene ABCG2 in PDAC cells by enhancing the methylation of hnRNPA1 at the R31 residue that, in turn, increases the binding of hnRNPA1 to ABCG2 mRNA. The binding of hnRNA1 to ABCG2 mRNA facilitates its export to the cytoplasm and enhances its expression level, thereby causing chemo-resistance in PDAC cells [112]. PRMT3 overexpression is also associated with an increased cell proliferation and anchorage-independent cell growth. These observations indicate that inhibition of PRMT3 is a new therapeutic strategy for chemo-resistant PDAC [112]. PRMT3 has been particularly explored for its role in regulating metabolic processes in PDAC cells. This shows that PRMT3 reprograms the metabolic process in PDAC cells via arginine methylation of GAPDH at the R248 residue [109]. This arginine methylation of GAPDH by PRMT3 enhances its catalytic activity, likely due to an enhanced formation of the active tetramer of GAPDH. Thus, overexpression of PRMT3 triggers metabolic reprogramming and enhances glycolysis and mitochondrial respiration in a GAPDH-dependent manner. Consequentially, PRMT3 overexpression sensitizes PDAC cells to the GAPDH inhibitor heptelidic acid [109].

### 3.3. PRMT5

PRMT5 belongs to the Type II PRMTs and is a major producer of SDMA in histone as well as non-histone proteins. It interacts with methylosome protein 50 (MEP50) to form a heterooctametric complex, eliciting its methyltransferase activity [58]. The expression level of both PRMT5 and MEP50 is often elevated in human cancer [113]. Overexpression of PRMT5 is observed at both mRNA and protein levels in PDAC tissues, which is associated with poor prognosis of PDAC patients [114,115,116].

The interaction of PRMT5 with cMYC oncogenic signaling in PDAC cells has been well documented. PRMT5 is found to play a critical role in glycolysis and tumorigenesis of PDAC cells via interacting with the F-box/WD repeat-containing protein 7 (FBW7)/cMyc axis [116]. Knockdown of PRMT5 in the pancreatic cancer cell lines MIA-PaCa 2 and SW1990 reduces the viability and colony formation capacity of the cells in vitro and the tumor volume in xenograft nude mouse models. Knockdown of PRMT5 in PDAC cells also inhibits glucose uptake and reduces lactate production. The uptake of 18F-FDG, an indicator of glucose uptake, is higher in subcutaneous tumors, while knockdown of PRMT5 reduces 18F-FDG uptake in these tumors. It turns out that PRMT5 regulates the expression of cMyc at the post-transcriptional level by inhibiting the E3 protein ligase FBW7 [117]. Knockdown of PRMT5 in PDAC cells reduces the protein level of cMYC without affecting its mRNA levels, causing an increased degradation of cMyc via the proteasomal degradation pathway facilitated by FBW7. The expression of FBW7, a tumor suppressor [117], is often reduced in human cancer cells [118]; however, knockdown of PRMT5 in PDAC cells elevated its expression, indicating that PRMT5 regulates FBW7 expression. The suppression of FBW7 expression by PRMT5 is primarily mediated via epigenetic modifications [116]. Thus, PRMT5 stabilizes cMYC protein by suppressing FBW7 expression, thereby promoting PDAC tumorigenesis.

A recent study also shows the connection of cMYC with PRMT5 in PDAC model systems. This study utilized an unbiased pharmacological screening approach in PDAC cells to identify the cMYC-associated epigenetic dependency [119]. PRMT5 inhibitors (PRMT5i) are identified as significant screening hits, where the sensitivity/efficacy of PRMT5i treatment in PDAC cells is directly associated with cMYC overexpression. There is a positive correlation between mRNA expression of PRMT5 and cMYC, and the overexpression of cMYC in patient-derived PDAC tumors is associated with high sensitivity towards PRMT5 inhibition. Evidently, PRMT5i treatment results in lower survival rate for PDAC cells with high cMYC expression (HUPT3, PaTu8988T, PSN1, and DanG) than those with lower cMYC expression (Panc1, PaTu8988S, HPAC, and Panc0504). Apoptosis seemed to be the primary mechanism by which the PRMT5i elicits therapeutic response in cMYC-expressing cells [119].

PRMT5 has been shown to facilitate EMT in PDAC cells via the EGFR/AKT/β-catenin pathway [120], indicative of its involvement in tumor metastasis. In the SW1990 and PaTu8988 cell lines, knockdown of PRMT5 reduces cell proliferation and colony formation, which is rescued upon ectopic re-expression of PRMT5. This is also evident in the xenograft mouse models, in which PRMT5 knockdown reduces the volume of the xenograft tumors. Consistently, the inhibition of the migration and invasion of PDAC cells is observed upon PRMT5 knockdown using the transwell migration and transwell invasion assays, which are rescued by ectopic re-expression of PRMT5. In addition, knockdown of PRMT5 results in an increased expression of the epithelial marker E-cadherin (at both mRNA and protein levels) and decreased expression of the mesenchymal markers Vimentin, Collagen I, and β-catenin, indicating that PRMT5 is involved in tumor metastasis. Mechanistically, knockdown of PRMT5 decreases the phosphorylation level of EGFR at Y1068 and Y1172, indicating that the arginine methylation of EGFR is essential for its phosphorylation in the Y1068 and Y1172 residues, without which the downstream signaling involving phosphorylation and activation of AKT/GSK3β and β-catenin is impaired. Note that arginine methylation of EGFR at the R1175 residue by PRMT5 has been shown to impact downstream signaling [121]. Thus, overexpression of PRMT5 in PDAC cells facilitates tumor EMT, thereby promoting tumor invasion and metastasis.

It is interesting to observe that PRMT5 inhibition affects cell proliferation which is synergized with the loss of Type I PRMTs, particularly PRMT1 [122]. The loss of PRMT1 in MIA PaCa-2 cells makes it highly sensitive towards the treatment with the PRMT5 inhibitor EPZ015666. The synergistic effect is also observed with the use of PRMT1 inhibitor MS023 in combination with the PRMT5 inhibitor EPZ015666. This suggests an overlapped spectrum of substrates of PRMT1 and PRMT5 in the cell proliferation pathway. Furthermore, inhibition of CARM1 or PRMT6 in combination with PRMT5 inhibition also shows similar consequences in these cells, but the effect is less significant compared with the inhibition of PRMT1, suggesting that PRMT1 is a primary Type I PRMT to compensate for the loss of PRMT5 in PDAC cells.

The use of PRMT5 inhibitors along with the chemotherapeutic agent gemcitabine has been shown to have a synergistic effect in tumor growth inhibition in PDAC cells through enhanced DNA damage [68]. Using an in vivo CRISPR gene knockout screening approach to search for the combinatorial targets of gemcitabine in PDAC, PRMT5 is identified as a druggable candidate that may act in synergy with gemcitabine to kill PDAC cells. In both in vitro and in vivo model systems, knockdown of PRMT5 or the use of PRMT5 inhibitors causes excessive DNA damage in PDAC cells when combined with gemcitabine treatment, a synergistic effect likely mediated by RPA exhaustion [68].

However, not all PDAC cells are equally sensitive to PRMT5 inhibition. PRMT5 inhibition appears to be more lethal in PDAC cells with the deletion of the tumor suppressor CDK2NA in the chromosome 9p21 locus. This is because the CDK2NA gene deletion is often associated with co-deletion of its adjacent genes in the genome, and one of the genes is the methylthioadenosine phosphorylase gene (MTAP) [123]. MTAP is essential for the metabolism of its substrate 5′-methylthioadenosine (MTA) to generate methionine and adenosine [124]. In MTAP-deleted cancer cells, the level of MTA is elevated which inhibits the methyltransferase activity of PRMT5 towards all of its substrates, increasing the sensitivity of cancer cells to further PRMT5 inhibition [125]. Therefore, in the MTAP deleted tumors, PRMT5 is a preferred molecular target for therapeutic development, because its inhibition appears to be more lethal than in tumor cells harboring the wild type MTAP gene. This is confirmed using a PDAC patient-derived organoids (PDOs) model with tumors derived from the pancreas that harbored the MTAP gene deletion. Treatment with the PRMT5-specific inhibitor EPZ015556 inhibits growth of the PDOs with the MTAP gene deletion, and this inhibition is significantly reduced in PDOs without the MTAP gene deletion [51].

## 4. Inhibitors of PRMTs

Since there is no protein arginine demethylase that has been consensually established to this date, arginine methylation is considered a relatively stable PTM that affects several downstream signaling cascades in cancer cells [126]. Given the role of PRMTs in the tumorigenesis and progression of human cancers [96,97,98,99,113], including PDAC [114,115,116], extensive effort has been directed towards the identification, synthesis, and application of PRMT inhibitors as potential cancer therapeutics. Indeed, PRMT inhibitors, especially inhibitors for PRMT1 and PRMT5 due to their established involvement in cancer progression, are currently being explored as therapeutics for hematological malignancies and solid tumors and have entered clinical trials (www.clinicaltrails.gov, accessed on 16 February 2024). The anticancer action of the major inhibitors against PRMT1, PRMT3, and PRMT5 is shown in Figure 3. Note that the cytotoxicity of PRMT inhibitors, especially inhibitors for PRMT5, are shown to be more specific towards cancer cells because of PRMT overexpression in cancer tissues and the reliance of tumor cells on PRMT activity [127]. Some of the recent clinical trials are listed in Table 4.

With the availability of assays to analyze PRMT activity, several groups of small molecules that target PRMTs have been developed, and these inhibitors elicit either specific or non-specific inhibition of different types of PRMTs. Typically, radiometric assays and antibody-based assays are utilized to study PRMT activity upon treatment with prospective small-molecular inhibitors in cellular model systems [128]. The mechanism of action for these small-molecule inhibitors are based on their ability to inhibit the methyltransferase activity of one or multiple PRMTs, mainly by inhibiting the binding of the PRMTs to their substrates or by occupying the SAM binding pockets in the PRMT enzymes, thereby diminishing methyltransferase activity [128]. While the mechanisms of action of most of these inhibitors are clearly described, the downstream effect they exert upon methyltransferase activity inhibition is a challenging task to elucidate because of the broad range of cellular activities mediated by PRMTs [128].

The AMI series of compounds are the first class of PRMT small-molecule inhibitors identified via ELISA-based high-throughput screening that shows selectivity towards Type I PRMTs and specificity in inhibiting arginine, but not lysine methyltransferase activity [129]. Following the identification of AMI compounds, effort in virtual and experimental screening has continued to uncover more PRMT1 inhibitors with high specificity [130]. A series of 1-substituted 1H-tetrazole derivative compounds have been screened for their PRMT1 inhibiting activity and a compound 9a has been deemed most potent, which selectively inhibits the methyltransferase activity of PRMT1 via interfering with the substrate binding site, demonstrated by molecular dynamics simulation. The inhibition of PRMT1 by compound 9a significantly reduces the cellular ADMA level and downregulates the Wnt/b-catenin signaling pathway in MDA-MB-231 cells [131]. MS023, also a Type I PRMT inhibitor, has been identified, displaying high potency to inhibit the activity of PRMT1, -3, -4, -6, and -8, while being completely inactive on the activity of Type II and III PRMTs [132]. A new Type I PRMT inhibitor, GSK3368715, is described as a potent and reversible inhibitor that inhibits all Type I PRMTs except PRMT3 [133]. GSK3368715 has been shown to have anticancer activity and has entered into a Phase I Clinical trial for the treatment of diffuse large B-cell lymphoma and selected solid tumors with MTAP deficiency (NCT03666988). However, this clinical trial has been terminated due to the lack of clinical efficacy [134]. Other inhibitors for Type I PRMTs, such as MS049, a dual inhibitor of CARM1 and PRMT6 [135], and SGC6870, a highly selective inhibitor of PRMT6 [136], have also been developed.

The development of inhibitors for Type II PRMTs, especially for PRMT5, has been quite successful. CMP5 is the first PRMT5-specific inhibitor developed by screening the ChemBridge CNS-Set library of 10,000 small-molecule compounds [137]. Treatment with CMP5 selectively reduces the viability of tumor cells, suggesting that PRMT5 is an ideal therapeutic target against cancer [137]. Another PRMT5 inhibitor EPZ015666 has been identified by using a homogeneous time-resolved fluorescence assay to screen a diverse library containing 370,000 small molecules [138]. This PRMT5 inhibitor acts by disrupting the MEP50:PRMT5 complex that is absolutely crucial for the methyltransferase activity of the enzyme. EPZ015666 is the first orally bioavailable and highly selective inhibitor of PRMT5 with antiproliferative effects in both in vitro and in vivo model systems [138]. Several PRMT5 inhibitors have been further developed, which have shown anticancer activity and have entered into clinical trials. These include GSK3326595 (EPZ015938) for solid tumors and non-Hodgkin’s lymphoma (NHL) (NCT02783300), as well as Myelodysplastic Syndrome (MDS) and Acute Myeloid Leukemia (AML) (NCT03614728); JNJ-64619178 for advanced solid tumors, NHL, and low-risk MDS (NCT03573310); PRT543 for advanced solid tumors and hematologic malignancies (NCT03886831); and PRT811 for advanced solid tumors, CNS lymphoma and Gliomas (NCT04089449).

Other PRMT inhibitors, such as EPZ020411 to inhibit PRMT6 by occupying its arginine binding site [139]; Compound II757, a pan-inhibitor for PRMTs [140]; and SGC3027, a potent PRMT7 inhibitor [141], have recently been described. A detailed patent review on PRMT inhibitors, especially inhibitors for PRMT1 and PRMT5, has been recently published [142].

The use of PRMT inhibitors in combination with the existing chemotherapeutics has proven to be beneficial in different cancer types. Combination of the PRMT5 inhibitor EPZ015666 with gemcitabine potentiates the DNA-damaging effect of gemcitabine in PDAC cells and helps to overcome therapeutic resistance via the HDR [68]. Treatment of breast cancer cells (MCF7, T-47D, MDA-MB-231, BT-549, and MDA-MB-468) with the combination of the PRMT5 inhibitor EPZ015666 and chemotherapeutic agents Etoposide/cisplatin demonstrates synergistic effect on the viability of these cells [143]. Notably, the immunotherapeutic efficacy of the anti-PD-L1 mAb in PDAC is enhanced when combined with the PRMT1 inhibitor PT1001B in a mouse model [104]. Some of the PRMT small-molecular inhibitors are orally bioavailable compounds, which do not exert extreme systemic toxicity. They have been reported to work well in combination with standard chemotherapeutics and are implemented in several clinical trials as a monotherapy or in combination (Table 4). Thus, these PRMT inhibitors are attractive therapeutic drug candidates for PDAC, where the major challenges are therapeutic resistance and severe toxicity due to multiple chemotherapeutics utilized during different stages of the treatment course.

## 5. Perspectives

The arginine methylation of histone and non-histone proteins has not been explored in depth. However, arginine methylation is a vital PTM in eukaryotic cells, where a large number of substrates rely on this PTM to elicit physiological activity. Protein arginine methylation plays critical roles in the initiation and progression of malignant diseases, including PDAC; therefore, targeting PRMTs is a logical strategy for cancer therapeutic development. While PRMT inhibitors have been tested in preclinical models for treatment of PDAC [68,104], clinical trials testing these inhibitors against PDAC have not been initiated. Based on the current status of PRMT inhibitors and the ongoing clinical trials, it is envisioned that new therapeutics targeting PRMTs, used along or in combination with other therapies, are likely to enter into clinical trials and become available in the clinical management of PDAC or other cancer patients in the future. Efforts on developing more enzyme-specific and potent PRMT inhibitors are expected to continue.

While studies using in vitro and in vivo model systems have shown the potential of PRMTs as therapeutic targets, elucidating the mechanism of action for these inhibitors has been a major challenge due to the substrate diversity and the cellular pathways involved. Furthermore, one could imagine that the arginine methylation site in any protein could vary in different cell types, depending on the expression and activity of individual PRMTs in the cell systems. For example, AKT has been recently identified to be methylated by PRMT5 at the R391 residue in MCF7 cells (SDMA formation) [143]. However, in the case of neuroblastoma cells, AKT is observed to be methylated at the R15 residue [144]. These cell type-dependent arginine methylation patterns indicate the intricacy in elucidating the cellular mechanisms related to arginine methylation across different types of cells and tumors. This intricacy has to be considered in our pursuit to better understand PRMT-mediated cellular processes.

Several studies have shown the existence of PRMT5 protein and mRNA in PDAC cell-derived exosomes, a group of small extracellular vesicles (EV) that mediate the transfer and function of biomolecules, including proteins, lipids, and nucleotides [145,146]. Given the involvement of exosomes in intercellular communication and cancer metastasis [147,148], the impact of PRMT5 on cell-to-cell communication via transfer of exosomes needs to be explored. In fact, one of the recent studies has indicated the impact of PRMT5 knockdown on the EV-associated pathways [149]. Expression of the proteins associated with EV biogenesis is upregulated upon PRMT5 knockdown in AML cells. This shows a biological relevance of PRMT5 in EV biogenesis that merits further investigation. In addition, one of the well-known biochemical functions of arginine methylation is in mediating protein phase separation [150,151,152]. A recent study has demonstrated the role of phase separation in the selective miRNA enrichment in exosomes [153]. This particular area needs to be further explored to establish the link between PRMTs and the biology of EVs. Furthermore, expression of PRMTs is often upregulated in cancer cells and tissues, resulting in different arginine methylation patterns of cellular proteins in normal versus cancerous cells or tissues. The different arginine methylation patterns may serve as biomarkers for early detection or disease monitoring for various malignancies, including PDAC. In particular, since tumor exosomes have been shown to be released into the circulation [154,155], arginine methylation patterns in plasma exosomes are likely indicators of PDAC [156], a lethal malignancy that desperately needs non-invasive biomarkers for early detection.

## 6. Conclusions

Protein arginine methylation is an important PTM in eukaryotic cells. Expression of PRMTs is often upregulated in PDAC cells and tissues, which facilitates changes in the transcriptional landscape, metabolic processes, EMT, and DNA damage responses, thereby promoting chemo-resistance and tumor progression (Figure 4). Due to the complications of current chemotherapeutics, such as severe systemic toxicity and therapeutic resistance, and the unsatisfactory outcome of immunotherapy and targeted therapy against PDAC, new therapeutic strategies are urgently needed in the clinical management of PDAC to improve patient survival outcomes. Targeting PRMT enzymes using small-molecule inhibitors provides a promising new line of therapy for PDAC. Studies have shown promising anticancer effects when combining PRMT inhibitors with existing chemotherapeutics to reduce tumor burden. Several PRMT inhibitors are now in clinical trials for both solid and hematological malignancies, either as a monotherapy or in combination. These clinical trials may help generate a new generation of therapeutics that can be clinically utilized for PDAC as well as other malignancies.

## Figures and Tables

**Figure 1 ijms-25-03958-f001:**
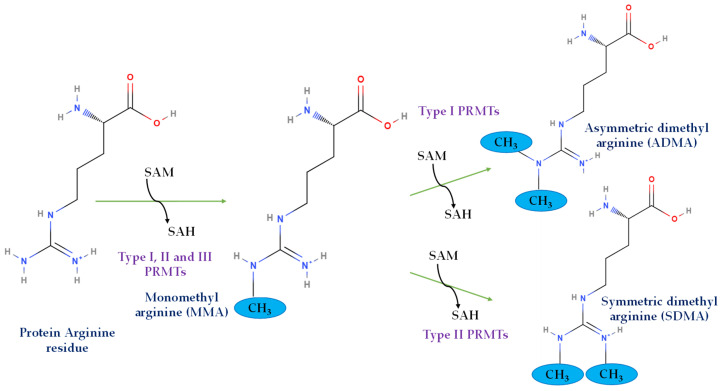
Different types of protein arginine methylation by PRMTs.

**Figure 2 ijms-25-03958-f002:**
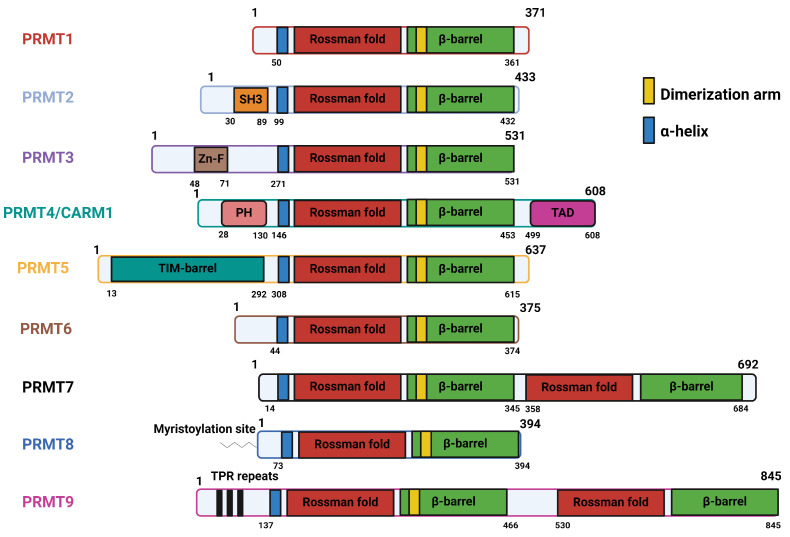
Structure of PRMT proteins (see Table 2 for structure features). Created with Biorender.com.

**Figure 3 ijms-25-03958-f003:**
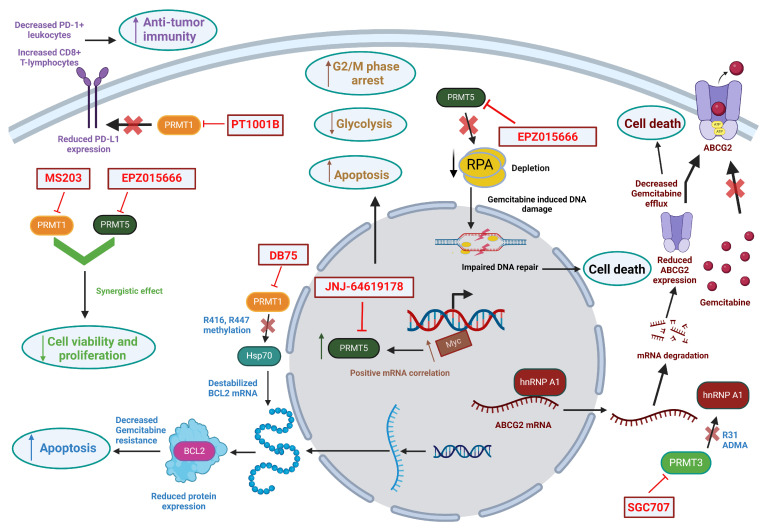
Anticancer action of PRMT inhibitors. Created with Biorender.com.

**Figure 4 ijms-25-03958-f004:**
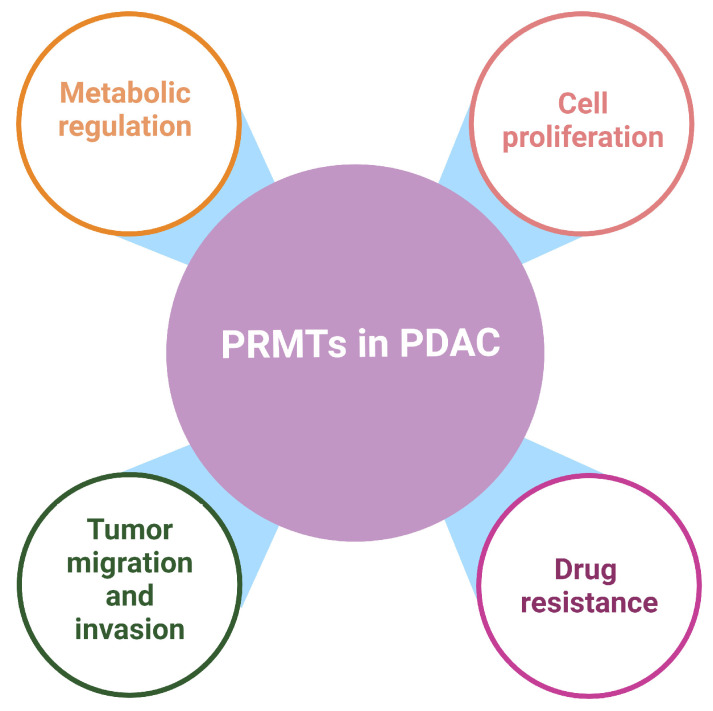
Role of PRMTs in pancreatic ductal adenocarcinoma (PDAC). Created with Biorender.com.

**Table 1 ijms-25-03958-t001:** Expression of PRMTs in the endocrine and exocrine compartments of the pancreas.

PRMTs	Endocrine Region	Exocrine Region
PRMT1	Medium	Low
PRMT2	Low	Medium
PRMT3	Medium	High to medium
PRMT4	Low	Medium to low
PRMT5	Low	Low to medium
PRMT6	Medium	High
PRMT7	Medium	Medium
PRMT9	Negative	Low

**Table 2 ijms-25-03958-t002:** Structural features of PRMTs.

Enzyme	Structural Features
PRMT1	Contain three canonical domains: N-terminal methyltransferase domain (Rossman fold), containing AdoMet binding pocket C-terminal ß-barrel domain, forming cylindrical structure corresponding to the arginine-substrate binding sites α-helical dimerization arm, an N-terminal part of the ß-barrel domain [53]
PRMT2	Three canonical domains, along with a unique Src homology 3 domain (SH3 domain) towards N-terminal extremity [54]
PRMT3	Three canonical domains, along with a unique C_2_H_2_Zn finger domain at the N-terminus for substrate binding [55,56]
PRMT4/CARM1	Three canonical domains, along with a C-terminal TAD domain and PH-like homology at the N-terminus for substrate recognition [57]
PRMT5	Three canonical domains, with N-terminal TIM barrel, which is essential for formation of complex with MEP50 [58]
PRMT6	Three canonical domains; no unique feature [59]
PRMT7	Three canonical domains, with two tandem methyltransferase domains due to gene duplication [38]
PRMT8	Three canonical domains, with a myristoylation site at the N-terminus that mediates its anchorage to the plasma membrane [36]
PRMT9	Three canonical domains, with two tandem methyltransferase domains and N-terminal TPR repeats [39]

**Table 3 ijms-25-03958-t003:** Subcellular localization and motif preferences in PRMTs.

PRMT	Cellular Localization *	Enzyme Type	Methylation Product	Motif Preference
PRMT1	**Cytoplasm,** Nucleus	I	MMA/ADMA	RGG or RxR
PRMT2	**Nucleus,** Cytoplasm	I	MMA/ADMA	RGG/RG
PRMT3	**Cytoplasm,** Nucleus	I	MMA/ADMA	RGG/RG
PRMT4/CARM1	**Nucleus,** Cytoplasm	I	MMA/ADMA	PGM
PRMT5	**Cytoplasm,** Nucleus	II	MMA/SDMA	GRG, PGM
PRMT6	**Nucleus,** Cytoplasm	I	MMA/ADMA	RxR
PRMT7	**Cytoplasm,** Nucleus	III	MMA	RxR
PRMT8	**Plasma** **membrane**	I	MMA/ADMA	RGG or RxR
PRMT9	**Cytoplasm,** Nucleus	II	MMA/SDMA	GAR

* Bold font indicates the dominant localization.

**Table 4 ijms-25-03958-t004:** Inhibitors of PRMTs currently in clinical trials.

Inhibitor	Target	Clinical Trial ID	Phase	Tumor
GSK3368715	Type I PRMT	NCT03666988	Phase I	Solid Tumors and Diffuse Large B-cell Lymphoma
AMG 193	PRMT5	NCT05094336	Phase I/II	MTAP-null solid tumors
JNJ-64619178	PRMT5	NCT03573310	Phase I	Advanced solid tumors, B cell non-Hodgkin lymphoma (NHL)
PF-06939999	PRMT5	NCT03854227	Phase I	Advanced solid tumors
PRT543	PRMT5	NCT03886831	Phase I	Relapsed or refractory solid tumors, lymphoma, and leukemia
PRT811	PRMT5	NCT04089449	Phase I	High grade gliomas, anaplastic astrocytoma, and advanced solid tumors
GSK3326595/EPZ015666	PRMT5	NCT04676516 NCT02783300 NCT03614728	Phase I/II Phase I Phase I/II	Early-stage breast cancer Advanced solid tumors, NHL Chronic myelomonocytic leukemia, Adult acute myeloid leukemia

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
