# Peer review of "Protein Arginine Methyltransferases in Pancreatic Ductal Adenocarcinoma: New Molecular Targets for Therapy"

_ijms, 2024, doi:10.3390/ijms25073958_

Round 1

Reviewer 1 Report

Comments and Suggestions for Authors

Abstract

Says: under cancer clinical trials

Suggestion:  under clinical trials

Says: and have the potential to become new generation of cancer therapeutics.

Suggestion: that can potentially become a new generation of anti-cancer drugs.

1.- Modify title to

1.- Introduction: Pancreatic Ductal Adenocarcinoma and the Need for New Therapeutics

Says: major type of pancreatic neoplasm of ductal origin

Suggestion: modify this sentence. PDAC can be originated also from acinar cells.

Says: Approximately 80-85% of the patients are diagnosed with PDAC when the disease has already metastasized, making them ineligible for surgical resection

Suggestion: Approximately 80-85% of the patients are diagnosed with PDAC when the disease has already metastasized or became locally advanced, making them ineligible for surgical resection

Says: , 3 out of 4 patients develop recurrence

Suggestion: , 3 out of 4 patients will develop a relapse

Says: they all have to undergo intensive chemotherapy

Suggestion: they usually all undergo…

Says FLORFIRINOX

Change for FORFIRINOX

Says: which is the combination

Suggestion: which is the association

Says: Irinotecan in adjunct with 5-Fluorouracil

Suggestion: Irinotecan with 5-Fluorouracil  

Says: if their disease is progressing during

Suggestion: if their disease progresses during

Line 47 correct FLORFIRINOX

Says: However, while the mutated KRAS protein is a “drug- gable” target using KRAS specific small molecule inhibitors

Suggest: However, while the mutated KRAS protein is a “drug gable” target using KRAS specific small molecule inhibitors (there are no FDA approved KRAS inhibitors for PDAC as yet)

Says: TP53, CDKN2A and SMAD4, also frequently occur in PDAC cells, which drive tumor progression independent of KRAS

Comment: I am not sure that TP53, CDKN2A and SMAD can drive tumor progression without an activated KRAS. While KRAS is an oncogene, the other three are tumor suppressor genes. Please add some more convincing references. I do not think Reference 14, Halbrook et al. is adequate for  this.

To the best of my knowledge for PDAC progression you need an active (mutated) KRAS or EGFR or PDGFR or FGFR.  Suppression of tumor suppressor genes is insufficient for PDAC. (This is a personal point of view).

Says: extracellular matrix (ECM) barrier, which contributes to the desmoplastic TME

Comment: the desmoplastic reaction IS the extracellular matrix barrier. Please modify the sentence.

Says: Recent advancement

Suggest: Recent advances

2. Arginine Methylation and PRMTs

Says: PRMTs are the writers of arginine

Question: what do you mean by writers? May be you want to say that PRMTs are the main arginine methylators?

Says: both nuclear as well as cytosolic compartment

Suggestion: both nuclear as well as cytosolic compartments

Says: While type I PRMTs are capable of forming

Suggestion: While type I PRMTs can form

Say: and type III PRMTs arecapable of forming mono-methyl arginine (MMA) only

Suggestion: and type III PRMTs can only produce mono-methyl arginine (MMA).

Change this sentence: The mechanism of arginine methylation by PRMTs is well described

Says: guanidino nitrogen atoms

Suggestion: guanidine nitrogen moieties

Says: All of the

Suggestion: All  the

2.1. Structural Basis, Localization and Motif Preference of PRMTs

Suggestion: A figure showing the PRMTs structure would be convenient for the reader.

Says: have high likelihood of  getting methylated

Suggestion: have high likelihood of being methylated  

Says: with the SAM as a methyl donor to methylate their substrates.

Suggestion: with  SAM as a methyl donor.

Says: majority of findings suggest

Suggestion: most findings suggest

2.2. Physiological Role of PRMTs

Says: therefore plays significant role

Suggestion: therefore plays a significant role

Says: as the writers of arginine  methylation,

Comment: I do not understand what you want to say with the word writers. Please, be more clear on this point.

Says: , such as estrogen receptor

Comment: which estrogen receptor?

Says: histone proteins is able to

Suggestion: histone proteins can

Says: transcriptional coactivator

Suggestion: transcriptional co-activator

Says: Studies have revealed that some of the PRMTs, especially PRMT1, PRMT3, and PRMT5, are inti-  mately associated with PDAC tumorigenesis, metastasis, and chemo-resistance.

Suggestion: It has been found that…

Comment: this sentence requires a reference.

Says: involving 90 patient tissue samples

Suggestion: involving tissue samples of 90 patients

Says: tissues compared with the adjacent normal

Suggestion: tissues compared to the adjacent normal

Says: and renders PDAC cells chemo-resistance

Suggestion: and renders PDAC cells chemo-resistant

Says: Studies have shown that PRMT3 upregulates expression of the multidrug resistant gene ABCG2 in PDAC cells, by enhancing the methylation of hnRNPA1 at the R31 residue that in turn increases the binding of hnRNPA1 to ABCG2 mRNA. The binding of hnRNA1 to ABCG2 mRNA facilitates its export to the cytoplasm and enhances its expression level, thereby causing chemo-resistance of PDAC cells.

Suggestion: this is an important sentence and needs a reference.

Says: It turns out that PRMT5 regulates the expression of cMyc at the post-transcriptional level by inhibiting the E3 protein ligase FBW7. Knockdown of PRMT5 in PDAC cells reduces the protein level of cMYC,  without affecting its mRNA levels, attributing to an increased degradation of cMyc via the  proteasomal degradation pathway facilitated by FBW7

Comment: this phrase requires a reference.

Says:  the CDK2NA gene deletion often  goes with co-deletion of its adjacent genes

Suggestion:  the CDK2NA gene deletion is often associated with co-deletion of its adjacent genes  

Says: heightening the sensitivity of cancer

Suggestion: increasing the sensitivity of cancer

5. Perspectives

Says: The arginine methylation of histone and non-histone proteins is not explored as well  as other PTMs such as phosphorylation and glycosylation

Suggestion: The arginine methylation of histone and non-histone proteins has not been explored in depth.  

Comments on the Quality of English Language

It needs some improvements

Author Response

We thank this reviewer for his/her thoroughness in reviewing our manuscript. We have addressed all concerns of this reviewer (see track-changed in the revision), and added a figure (Figure 2) to show the structure of PRMTs according to the suggestions.

Reviewer 2 Report

Comments and Suggestions for Authors

This is a very well written review that addresses the role of PRMTs in pancreatic cancer and also addresses the development of anti-cancer therapeutics based on these findings. The authors systematically address the role of different PRMTs in PDAC, starting with arginine metylation, then discussing different PRMT5s, and finally ending with the developed pharmacological inhibitors.

I have few concerns at the moment:

1) The authors could discuss the effects of PRMT inhibitors on normal cells in terms of clinical trials and in the context of normal pancreatic cells.

2) Most of the inhibitors in clinical trials are against PRMT5, only one is against PRMT1, and all other PRMTs are not targeted in clinical trials-- this can be discussed.

3) All clinical trials of PRMT inhibitors are not  taking place in PDAC - the authors discuss some work in PDAC cell lines using combination therapies, but some more discussion of this issue will be beneficial.

4) I miss the summarizing signaling schemes showing how application of a particular inhibitor leads to apoptosis induction or inhibition of DNA repair or cell proliferation via inhibition of a particular PRMT enzyme. These need to be added to this review.

Author Response

We appreciate this reviewer’s positive comments and suggestions.

  1. The authors could discuss the effects of PRMT inhibitors on normal cells in terms of clinical trials and in the context of normal pancreatic cells.

Response: We have discussed the selective action of PRMT inhibitors towards cancer cells and added a corresponding reference (page 12/25, line 410-413).

  1. Most of the inhibitors in clinical trials are against PRMT5, only one is against PRMT1, and all other PRMTs are not targeted in clinical trials-- this can be discussed.

Response: We have added discussion on page 11/25, line 406.

  1. All clinical trials of PRMT inhibitors are not  taking place in PDAC - the authors discuss some work in PDAC cell lines using combination therapies, but some more discussion of this issue will be beneficial.

Response: We have added discussion on page 13, line 490-495.

  1. I miss the summarizing signaling schemes showing how application of a particular inhibitor leads to apoptosis induction or inhibition of DNA repair or cell proliferation via inhibition of a particular PRMT enzyme. These need to be added to this review.

Response: We have added a figure (Figure 3) to indicate the anticancer action of the major PRMT inhibitors.

Round 2

Reviewer 2 Report

Comments and Suggestions for Authors

excellent revisions: I do not have anymore comments